# The Role of SNHG15 in the Pathogenesis of Hepatocellular Carcinoma

**DOI:** 10.3390/jpm12050753

**Published:** 2022-05-06

**Authors:** Christos Damaskos, Nikolaos Garmpis, Dimitrios Dimitroulis, Anna Garmpi, Evangelos Diamantis, Panagiotis Sarantis, Vasiliki E. Georgakopoulou, Alexandros Patsouras, Markos Despotidis, Dionysios Prevezanos, Athanasios Syllaios, Georgios Marinos, Evangelos Koustas, Christos Vallilas, Efstathios A. Antoniou, Konstantinos Kontzoglou, Spyridon Savvanis, Gregory Kouraklis

**Affiliations:** 1Renal Transplantation Unit, Laiko General Hospital, 11527 Athens, Greece; prevedio@hotmail.com; 2N.S. Christeas Laboratory of Experimental Surgery and Surgical Research, Medical School, National and Kapodistrian University of Athens, 11527 Athens, Greece; nikosg22@hotmail.com (N.G.); efstathios.antoniou@gmail.com (E.A.A.); kckont@med.uoa.gr (K.K.); 3Second Department of Propedeutic Surgery, Laiko General Hospital, Medical School, National and Kapodistrian University of Athens, 11527 Athens, Greece; dimitroulisdimitrios@yahoo.com; 4First Department of Propedeutic Internal Medicine, Laiko General Hospital, Medical School, National and Kapodistrian University of Athens, 11527 Athens, Greece; annagar@windowslive.com; 5Endocrinology Unit, Academic Department of Internal Medicine, Agioi Anargyroi General Oncology Hospital, National and Kapodistrian University of Athens, 14564 Kifisia, Greece; vaggelisd01@gmail.com; 6Molecular Oncology Unit, Department of Biological Chemistry, Medical School, National and Kapodistrian University of Athens, 11527 Athens, Greece; psarantis@med.uoa.gr (P.S.); vang.koustas@gmail.com (E.K.); chris-vallilas@hotmail.com (C.V.); 7Department of Pulmonology, Laiko General Hospital, 11527 Athens, Greece; vaso_georgakopoulou@hotmail.com; 8Second Department of Pulmonology, Sotiria Hospital, 11527 Athens, Greece; patsouras.alexandros@gmail.com; 9First Department of Surgery, Laiko General Hospital, Medical School, National and Kapodistrian University of Athens, 11527 Athens, Greece; markosd1995@yahoo.gr (M.D.); nh_reas@hotmail.com (A.S.); 10Department of Hygiene, Epidemiology and Medical Statistics, Medical School, National and Kapodistrian University of Athens, 11527 Athens, Greece; gmarino@med.uoa.gr; 11Internal Medicine Department, Elpis General Hospital, 11522 Athens, Greece; savvanis@live.com; 12Medical School, National and Kapodistrian University of Athens, 11527 Athens, Greece; gkouraklis@hotmail.com

**Keywords:** SNHG15, lncRNA, hepatocellular, carcinoma, cancer, hepatocarcinogenesis

## Abstract

Long non-coding RNAs (lncRNAs) are transcripts of more than 200 nucleotides which cannot be translated into proteins. Small nucleolar RNA host gene 15 (SNHG15) is a lncRNA whose dysregulation has been found to have an important impact on carcinogenesis and affect the prognosis of cancer patients in various cancer types. Hepatocellular carcinoma (HCC) is one of the most common cancers with a poor long-term prognosis, while the best prognostic factor of the disease is its early diagnosis and surgery. Consequently, the investigation of the mechanisms of hepatocarcinogenesis, as well as the discovery of efficient molecular markers and therapeutic targets are of great significance. An extensive literature search was performed in MEDLINE in order to identify clinical studies that tried to reveal the role of SNHG15 in HCC. We used keywords such as ‘HCC’, ‘hepatocellular carcinoma’, ‘SNHG15’ and ‘clinical study’. Finally, we included four studies written in English, published during the period 2016–2021. It was revealed that SNHG15 is related to the appearance of HCC via different routes and its over-expression affects the overall survival of the patients. More assays are required in order to clarify the potential role of SNHG15 as a prognostic tool and therapeutic target in HCC.

## 1. Introduction

Hepatocellular carcinoma (HCC) is the major type of liver cancer, one of the most malignant cancers as it represents the fourth leading cause of cancer-related deaths worldwide, and one of the six most frequent cancers overall [1,2]. It has been proven that environmental and nutrition factors affect the appearance of HCC. More specifically, the risk factors for HCC include chronic HBV and HCV, chronic liver infections, autoimmune hepatitis, chronic alcohol use, obesity and diabetes mellitus [2,3]. Apart from the aforementioned factors, epigenetic alterations may also contribute to HCC development [4]. The clinical presentation of HCC varies from asymptomatic to a possibly acute life-threatening complication. Many patients who develop HCC have no symptoms that can be related to the tumor; especially those who have been diagnosed with HCC at an early stage [2]. Moreover, HCC seems to be strongly related to cirrhosis. This is probably caused by mechanisms that induce the selection of proliferative cells as a response to the reduced capacity of hepatocytes to proliferate and regenerate at the cirrhosis stage [5]. The diagnosis of HCC can be difficult and often requires the use of more than one imaging modality as well as biomarkers such as Alpha Fetoprotein (AFP). Nonetheless, the delayed detection of the disease, which results in poorer prognosis due to the inefficiency of therapeutic interventions for terminally staged tumors, remains a severe problem [6]. Considering that the surgical removal of the tumor is the best prognostic factor, the diagnosis in an early stage is quite important [7,8,9]. Additionally, orthotopic liver transplantation is a potentially curative treatment and the best treatment option for the patients in terminal stage. Liver transplantation is effective for the patients with HCC whose tumor is either solitary and not larger than 5 cm, or multifocal, up to three lesions with each lesion size of 3 cm or smaller. The 5-year overall survival rate of liver transplantation for HCC is 75% [10]. Thus, it is of great importance to identify prognostic and diagnostic factors at an early stage of HCC, giving the patients higher likelihood of survival.

The majority of the human transcribed genome consists of noncoding RNAs (ncRNAs). Noncoding RNAs include microRNAs (miRNAs), a class of short ncRNAs, and a variety of long ncRNAs (lncRNAs). Small nucleolar RNA host gene 15 (SNHG15) is a lncRNA. Among the characteristics of the lncRNAs are their size (longer than 200 nucleotides), their inability to code proteins, and therefore their identification as transcriptional “noise”. Moreover, they have an important role in several cellular processes, such as cell differentiation, cell cycle control and cancer-cell metastasis, and biological processes such as tissue differentiation and embryonic development [11,12,13,14,15]. There are also studies suggesting that lncRNAs contribute to carcinogenesis of various tumors by taking part in tumor suppressing pathways [16]. A lot of lncRNAs are abnormal in different types of cancer and their expressions are related to carcinogenesis [17]. For instance, studies that support the increase in gastric cancer by the over-expression of lncRNA H19 have been reported [18]. Moreover, in patients with HCC, the expression of lncRNA PVT1 showed a significant increase in HCC tissues, and it was indicative of the reappearance of this specific type of cancer [19]. As a result, lncRNAs may act as cancer biomarkers [20].

The lncRNA SNHG15 is an 837 bp long gene, located on chromosome 7p13, and is described as being up-regulated in gastric cancer tissue samples and cell lines [21]. When SNHG15 is produced in higher quantities, it is associated with tumorigenic functions, including proliferation and metastasis in a variety of cancer types [22,23,24]. SNHG15 has been linked to lower survival in a lot of human malignancies, including breast cancer, colorectal cancer, gastric cancer, glioma, hepatocellular cancer, lung cancer, osteosarcoma, ovarian cancer, pancreatic cancer, renal cell carcinoma and thyroid cancer [25,26]. In pancreatic cancer tissue samples and cell lines, its expression was higher than expected, suggesting that it has pro-oncogenic functions [27]. In addition, in breast cancer it has been proved that by sponging miR-211-3p, SNHG15 contributes to proliferation, migration and invasion [23] and by sponging miR-141, it promotes proliferation, invasion and autophagy in osteosarcoma cells [28]. However, the exact mechanisms of the oncogenic function of SNHG15 are not yet clear and further studies are needed so that we fully clarify its role.

Sorafenib, a multikinase inhibitor, has been the sole available treatment option for advanced HCC since 2007; as HCC has proven to be chemotherapy resistant despite the other targeted therapies (Lenvatinib, Cabozantinib, Regorafenib, Ramucirumab), immunotherapies (Pembrolizumab, Nivolumab) and combination therapies [29], the management of HCC remains challenging and so the interest of the researchers is turning to new possible therapeutic approaches. On the other hand, as aforementioned, SNHG15 seems to play an oncogenic role in the pathogenesis of several cancers. Thus, in this study, we focus on the role of SNHG15 on oncogenesis of HCC by reviewing the current literature regarding its potential biological and molecular mechanisms. In this way, we aim to contribute to the discussion about the potential role of SNHG15 as a diagnostic and prognostic biomarker, as well as a possible potential therapeutic target.

## 2. Materials and Methods

An extensive literature search was performed in MEDLINE (via PubMed) in order to identify clinical studies that tried to reveal the role of SNHG15 in HCC. Keywords such as ‘HCC’, ‘hepatocellular carcinoma’, ‘SNHG15’ and ‘clinical study’ were used in our search strategy. The selection of the articles was based on PRISMA guidelines. A total number of 304 publications were identified; after the removal of duplicates, 302 remained. Following screening, 298 articles were excluded because they did not fulfill the inclusion criteria regarding the aforementioned keywords. Finally, we included 4 studies written in English and published during the period 2016–2021, in which the resected HCC specimens were studied on histopathological and genetical level. The inclusion process according to PRISMA guidelines is demonstrated in Figure 1.

## 3. Results

### 3.1. SNHG15 Expression

A common result that emerged from all the analyzed studies was the fact that SNHG15 was over-expressed in HCC. This fact was apparent when pairs of healthy non-cancerous tissues and specimens of liver cancer were compared with the use of RT-qPCR, but it was also evident when HCC cell lines were compared with normal liver cell lines. Apart from this, the higher expression of SNHG15 was correlated with tumors with worse pathologic and prognostic characteristics. Among 201 paired specimens of HCC and normal liver tissue, and 269 total specimens of HCC, it was found that expression of SNHG15 was significantly higher in cancerous tissues [24,30,31,32]. Furthermore, the higher expression of SNHG15 was correlated with an advanced histologic grade, higher TNM stage, vein invasion and poorer overall survival. The correlation between SNHG15 expression and overall survival was also statistically significantly independent of the other factors [24].

Dai et al. showed similar results indicating an association between the expression of SNHG15 and tumor size, TNM stage and poor differentiation of HCC [30], while Chen et al. also confirmed that SNHG15 up-regulation affects tumor stage, TNM, differentiation of HCC, vascular invasion, invasive depth (T1–2 vs. T3–4) and five-year survival [32].

### 3.2. Proliferation, Migration and Invasion

Another common finding was the relationship between the up-regulation of SNHG15 and the proliferation, migration and invasion of HCC cells [30,31,32]. Dai et al. confirmed this finding by proving that knockdown of SNHG15 impeded proliferation, migration and invasion of HCC cells [30], while Ye et al. additionally found that apart from this, SNHG15 knockdown had a positive impact on G1/G0 phase cell cycle arrest and a negative one on the S phase arrest [31].

Last but not least, Chen et al. confirmed these findings in HCC cell lines as well as in vivo, by injecting in nude mice HCC cells transfected with plasmids with low and high expression of SNHG15. They found that SNHG15 knockdown impeded colony formation, invasion and migration of HCC cells as well as enhanced apoptosis of HCC cells, while the tumor size and volume was decreased in nude mice injected with HCC cells transfected with plasmids with low expression of SNHG15 [32].

### 3.3. Interaction with miRNAs

Trying to comprehend the molecular and biologic mechanisms with which SNHG15 plays such a role in tumorigenesis of HCC, the researchers found potential binding sites of SNHG15 with miRNAs [30,31,32]. Dai et al. found a lower level of miR-490-3p in HCC tissues in comparison with normal tissues. In HCC cells, the expression of SNHG15 had a reverse effect on miR-490-3p. Furthermore, up-regulation of SNHG15, and as a result down-regulation of miR-490-3p, led to up-regulation of histone deacetylase inhibitor 2 (HDAC-2). They also showed that there is a binding site between miR-490-3p and HDAC-2 and that HDAC-2 is increased in HCC. As a result, the authors suggested that SNHG15 controls miR-490-3p by sponging it and that the SNHG15/miR-490-3p axis contributes to tumorigenesis by controlling HDAC-2 [30].

On the other hand, Ye et al. found potential binding sites between SNHG15 and miR-141-3p. The miR-141-3p was found to be down-regulated in HCC tissues and it is known to regulate E2F3 and ZEB2 genes, which are up-regulated in HCC cancer. Consequently, it was proposed that up-regulation of SNHG15 sponges miR-141-3p and leads to over-expression of E2F3/ZEB2, contributing to tumorigenesis [31].

Chen et al. also found a potential binding site between SNHG15 and one miRNA. Specifically, it was found that SNHG15 could regulate miR-18b-5p-LIM-only-4 (LMO4) axis by binding to miR-18b-5p. Up-regulation of SNHG15 led to down-regulation of miR-18b-5p and up-regulation of LMO4. The miR-18b-5p is a known tumor suppressor, while LMO4 a known oncogene [32].

Table 1 summarizes all the aforementioned results of clinical trials on SNHG15 in HCC.

Figure 2 summarizes all the aforementioned effects of SNHG15 on HCC.

## 4. Discussion

lncRNAs have lately been appreciated as crucial modulators of various cellular processes through modification of gene expression, and thus their role as tumor suppressors or oncogenes has started to be explored. Apart from SNHG15, MNX1-AS1 is another lncRNA that seems to take part in tumorigenesis of HCC [33]. Moreover, recent research demonstrated that a combination of inhibitors of four protein kinases (RAS/RAF/MEK/ERK) with other pathway inhibitors that overexpressed in HCC could lead to satisfactory outcomes; overcoming the resistance mechanisms which are still unknown. Gnoni et al. described how any BRAF mutation in patients with advanced-stage HCC could be affected by lncRNA. As a result, lncRNA could play a crucial role in cell proliferation through the BRAF pathway and create combined targeted therapies [34]. In addition, RAS/RAF/MEK/ERK also regulate the MAPK/ERK pathway [35]. This pathway appears to be activated in half of patients of early-stage HCC and in nearly all the patients with advanced HCC [36]. Several molecular agents targeting MAPK/ERK signaling pathway have been studied recently apart from Sorafenib. Regofarinib and Cabozantinib, multi-kinase inhibitors, are indicated as second-line therapy leading to better overall survival compared to placebo treatment [37,38]. Monoclonal antibodies in conjunction with lncRNAs might expand the current therapeutic approaches and understand accurately the nature of HCC.

As concluded by our study, SNHG15 is over-expressed in HCC and also correlated to its histological grade, TNM stage, vein invasion and overall survival in HCC patients [24,30,31,32]. SNHG15 is located on 7p13 and it is known to play a critical role in several cancers as oncogene, and so it could be used as an adverse prognostic biomarker [26,39]. However, the further reveal of its function and of the mechanisms of its oncogene functions could lead to its use as a therapeutic target, too [32].

A hypothesis for the biologic function of lncRNAs, which could apply to SNHG15 and is also proved by the studies analyzed in this review, is the involvement of SNHG15 in competing endogenous RNA (ceRNA) networks. In these networks, lncRNAs can regulate miRNAs expression by binding to them and sponging them. These miRNAs, on the other hand, control expression of genes associated with tumorigenesis [10,30,31,32,40].

Dai et al. showed that SNHG15 up-regulation results in increased expression of HDAC-2 [30]. Human HDACs are divided into four groups (I, II, III and IV) and control several cellular procedures via deacetylation of histones and non-histone proteins [41]. Consequently, HDAC-2 has been found to participate in carcinogenesis of several tumors through epigenetic regulation of cell cycle, apoptosis and differentiation of cancerous cells. HDAC-2 has been found to be up-regulated in HCC, lung, pancreatic, colorectal and breast cancer and it has also been correlated to chemotherapy resistance [34,41,42,43,44,45,46].

Chen et al. concluded that up-regulation of SNHG15 leads to up-regulation of LMO4 [32]. LMO4 also belongs to a family of proteins that regulate transcription and they have been found to be related to several cancers such as leukemia, breast, and head and neck cancer [47,48].

Therefore, it is concluded that SNHG15 seems to participate in oncogenesis of HCC and plays an important role in its evolution. The hypothesis of ceRNAs seems also to be able to explain the biologic and molecular mechanisms of the oncogenic action of SNHG15. As a result, SNHG15 could be an ideal target for therapeutic actions and the evolution of new targeted therapies, as well as an ideal prognostic biomarker.

Our review tries to conclude the role of SNHG15 in carcinogenesis of HCC and its use as prognostic and therapeutic tool. The studies chosen for the analysis of our study included specimens of resected HCC as well as HCC and normal liver cell lines. One study also used nude mice to reveal the role of up-regulation of SNHG15 in the evolution of HCC in vivo [32]. As a result, our review can contribute critically to the discussion regarding the role and the possible clinical use of SNHG15 in HCC.

Certainly, the role of SNHG15 is still unclear and the research is in preliminary stages, but it has been described in a plethora of malignancies. In a vast part of them, it is overexpressed having an oncogenic role while in others, e.g., the thyroid cancer, is tumor suppressor (Table 2) [21,22,23,27,28,49,50,51,52,53,54]. Similar to HCC, SNHG15 elevation is strictly associated with worst prognosis, advanced-stage malignancy and possible metastasis proving to be a possible key-factor understanding the HCC nature and its oncogenic pathways. Nonetheless, the number of the studies included regarding the HCC is limited, so as a result, further research is needed in order to reconfirm such findings, especially in vivo.

## 5. Conclusions

Many different lncRNAs have the ability to control HCC progression [55]. Moreover, up-regulation of SNHG15 expression is closely associated with HCC development. Therefore, a new potential marker and target for HCC prognosis and treatment could be developed based on these results. More studies should be carried out in order to explore other miRNAs targeted by SNHG15 in HCC. However, the results described in this article offer significant insights concerning HCC and offer a basis for the development of novel therapeutic strategies for patients with HCC.

## Figures and Tables

**Figure 1 jpm-12-00753-f001:**
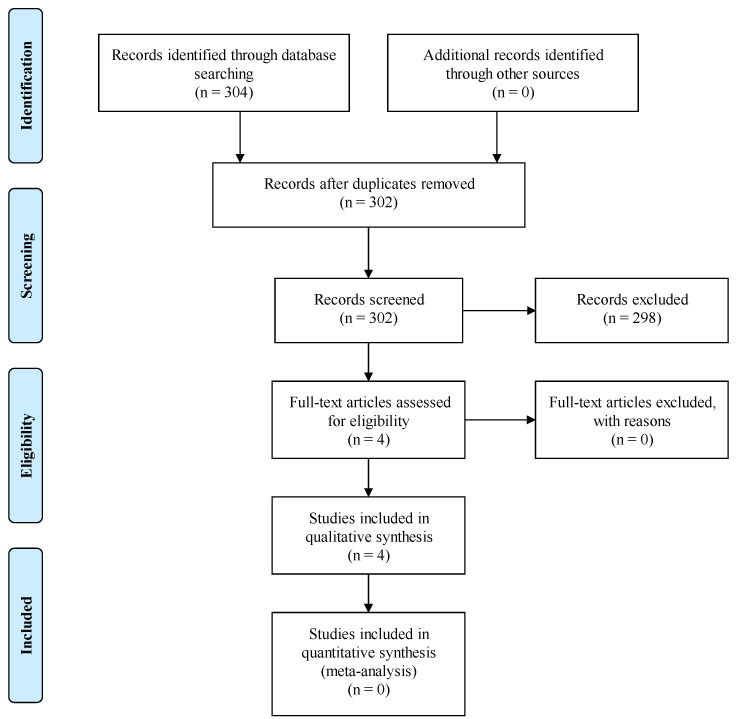
PRISMA flow diagram for the current article process.

**Figure 2 jpm-12-00753-f002:**
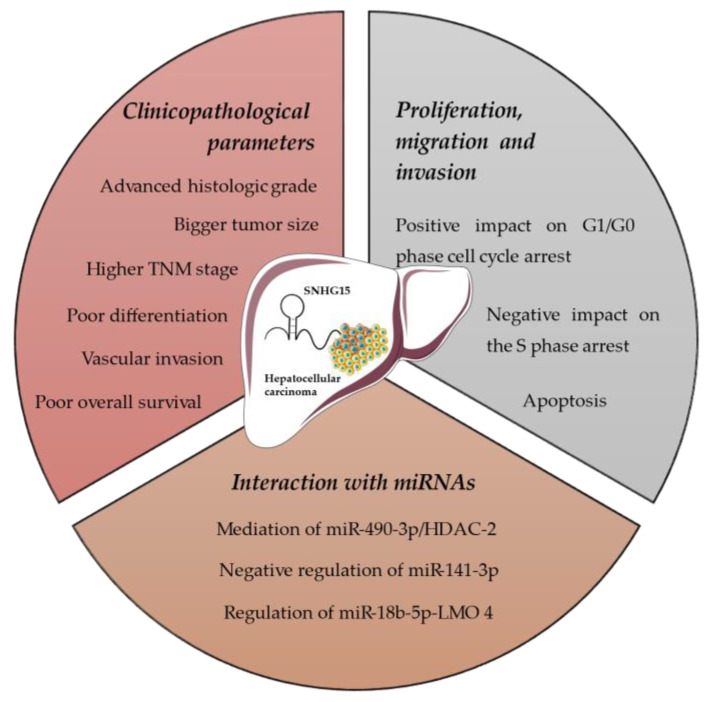
Effects of SNHG15 on hepatocellular carcinoma.

**Table 1 jpm-12-00753-t001:** Clinical trials on SNHG15 in hepatocellular carcinoma.

Study	Design	Material	Drug	Result
Zhang et al., 2016 [24]	Associations between		-	lncRNA SNHG15 may serve as an efficient clinical biomarker and a therapeutic target for HCC patients.
	clinicopathological parameters and lncRNA SNHG15 expression were evaluated using chi-square tests.	152 paired HCC tissues and adjacent normal tissues.		
Dai et al., 2019 [30]	- HCC cell lines HuH-1, HuH-7 and normal human liver cells L-O_2_. - The growth conditions of these cells are 5% CO_2_, 37 °C and culture solution. - si-SNHG15, pcDNA3.1-SNHG15 vectors, si-HDAC-2 and miR-490-3p mimics or inhibitor were transfected into HuH-1 or L-O2 cells. - Untreated cells were used as the controls.	101 HCC patients.	Lipofectamine 2000 (Invitrogen).	lncRNA RNA SNHG15 promotes HCC progression by mediating the miR-490-3p/HDAC-2 axis in HCC.
Ye et al., 2019 [31]	- 4 HCC cell lines and a normal human LO_2_ liver cell line. - siRNAs against si-SNHG15 and a non-targeting control (si-NC). - miR-141-3p, a non-targeting miR control (miR-NC), and the miR-141-3p inhibitor (miR-141-3p in).	58 paired HCC samples and adjacent matched adjacent normal tissues.	- Penicillin, streptomycin for cell culture in a humid 37 °C environment with 5% CO_2_ atmosphere. - Lipofectamine 2000.	SNHG15 promoted HCC progression throws negative regulation of miR-141-3p, identifying a potential novel HCC way of treatment.
Chen et al., 2021 [32]	- 4 human liver cancer cell lines and one normal liver cell line.		High-glucose Dulbecco’s modified eagle medium and RPMI 1640 medium (HyClnoe, Logan, UT), in 5% FBS-contained medium, detached with 0.25% trypsin and passaged, MTT solution, Dimethyl sulfoxide, Lipofectamine 2000 (Invitrogen, Carlsbad, CA).	- Up-regulation of SNHG15 in HCC cells. - Knockdown of SNHG15 impedes colony formation, invasion and migration of HCC cells and enhances apoptosis. - SNHG15 regulates miR-18b-3p-LMO4 axis.
	- Transfection with SNHG15 low-expression negative control (NC) plasmids, SNHG15 low expression plasmids, miR-18b-5p mimic NC, miR-18b-5p mimic, SNHG15 over-expression plasmids and mir-18b5p mimic NC, or SNHG15 over-expression plasmids and mir-18b-5p mimic. - Female BALB/c nude mice (*n* = 28, 4 weeks) were injected with the transfected cells.	58 paired HCC and normal tissue specimens.		

lncRNA: Long noncoding RNA; SNHG15: Small nucleolar RNA host gene 15; HCC: Hepatocellular carcinoma; HDAC: Histone deacetylase; RPMI: Roswell Park Memorial Institute; FBS: Fetal bovine serum; MTT: 3-(4, 5-dimethylthiazol-2-yl)-2, 5-diphenyltetrazolium bromide; LMO4: miR-18b-5p-LIM-only-4.

**Table 2 jpm-12-00753-t002:** Studies indicating SNHG15 expression in different cancer types and its effects.

Study	Type of Cancer	Role	Expression	Effect on Cancer Cells
Chen et al., 2016 [21]	Gastric	Oncogenic	Up-regulation	Proliferation, apoptosis, migration, invasion
Cui et al., 2016 [22]	Lung	Oncogenic	Up-regulation	Proliferation, apoptosis, migration, invasion
Ma et al., 2017 [27]	Pancreatic	Oncogenic	Up-regulation	Proliferation, apoptosis, cell cycle arrest
Ma et al., 2017 [49]	Glioma	Oncogenic	Up-regulation	Proliferation, migration, angiogenesis, tube formation
Liu et al., 2017 [28]	Osteosarcoma	Oncogenic	Up-regulation	Proliferation, migration, autophagy, invasion
Kong et al., 2018 [23]	Breast	Oncogenic	Up-regulation	Proliferation, apoptosis, migration, invasion
Du et al., 2018 [50]	Renal cell	Oncogenic	Up-regulation	Proliferation, migration, invasion, apoptosis, cell cycle arrest
Li et al., 2019 [51]	Colorectal	Oncogenic	Up-regulation	Proliferation, apoptosis, migration, invasion
Zhang et al., 2019 [52]	Prostate	Oncogenic	Up-regulation	Migration, invasion
Liu et al., 2019 [53]	Thyroid	Oncosuppressive	Down-regulation	Proliferation, migration, invasion
Qu et al., 2019 [54]	Ovarian	Oncogenic	Up-regulation	Proliferation, migration, invasion

## Data Availability

Not applicable.

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
