# Peer review of "The Role of SNHG15 in the Pathogenesis of Hepatocellular Carcinoma"

_jpm, 2022, doi:10.3390/jpm12050753_

Round 1
Reviewer 1 Report
Christos Damaskos et al. summarized the role of SNHG15 in the pathogenesis of hepatocellular carcinoma.
The following have not been addressed:
1) The rationale of why the authors came up with this review.
2) What is the information that is not exactly available that motivated the authors to come up with this information. What are the current caveats and how do the authors highlight the current research in answering them? If not they need to address in future directions.
3) This reviewer personally misses some important information regarding the few therapeutic strategies for advance HCC on poor knowledge of its biology. For several years, sorafenib, a tyrosine kinase inhibitors (TKI) inhibitor, has been the approved treatment option, to date, for advanced HCC patients. Its activity is the inhibition of the retrovirus-associated DNA sequences protein (RAS)/Rapidly Accelerated Fibrosarcoma protein (RAF)/mitogen-activated and extracellular-signal regulated kinase (MEK)/extracellular-signal regulated kinases (ERK) signaling pathway. However, the efficacy of sorafenib is limited by the development of drug resistance, and the major neuronal isoform of RAF, BRAF and MEK pathways play a critical and central role in HCC escape from TKIs activity. Advanced HCC patients with a BRAF mutation display a multifocal and/or more aggressive behavior with resistance to TKI. Moreover, also long non-coding RNA (lnc-RNA) have been studied in epigenetic studies for BRAF aggressiveness in HCC. So far, lnc-RNA of BRAF could be another mechanism of cancer proliferation and TKI escape in HCC and the inhibition could become a possible strategy treatment for HCC: please refer to PMID: 31766556 and expand.
4) In the frame of this thinking, a plethora of second-line therapies have been recently introduced for HCC treatment with promising results. A meta-analysis of second-line treatments for HCC has been performed to better tailor their use based on improved patient stratification and to identify the best available option (refer to PMID: 34146196). Nonetheless, studies focusing on epigenetic and lnc-RNA might expand the taloired approach and therapeutic landscape.
5) Does this role of endothelial cells in angiogenesis in a tumor micro-environment involve hypoxia and lnc-RNA? Since hypoxia is a key factor for angiogenesis, the authors need to substantiate.
6) The authors need to highlight what new information the review is providing to enhance the research in progress.
Author Response
Dear Reviewer
We agree with your comments. All the responses are highlighted with yellow. Thank you very much.
Reviewer 2 Report
In this manuscript, the authors summarized gene expression studies in HCC using clinical samples. The content is interesting, and the manuscript, itself, is well written. However, the number of main studies cited in this manuscript is only four. Meta-analyses were not performed, and the authors just introduce the results of each study. Thus, the clinical significance of SNHG15 seems to be overestimated. I would like to suggest the authors to collect and review not only clinical studies in terms of HCC, but also basic oncobiological studies on any cancers.
- (lines 107–108) The inclusion criteria should be defined.
- (lines 118–123) An appropriate reference should be cited for each sentence.
- (line 143) "colon" –> "colony"(?)
- Because this is the journal of "Personalized Medicine", the authors should describe, as perspective, clinical application of SNHG15 for treatment and diagnosis in more detail.
Author Response

(The authors gave the same response as above.)

Round 2
Reviewer 1 Report
The authors have clarified several of the questions I raised in my previous review. Most of the major problems have been addressed by this revision.
Reviewer 2 Report
Although the authors addressed some of my comments, the clinical significance of SNHG15 is still unclear and overestimated unless more additional research articles are included.